# Interactive Laboratories for Science Education: A Subjective Study and Systematic Literature Review

**Numan Ali** [1,2,*] **, Sehat Ullah** [1] **and Dawar Khan** [3,4]

1   Department of Computer Science & IT, University of Malakand, Chakdara 18800, Pakistan
2   Department of Computer Games Development, FCAI, Air University Islamabad, Islamabad 44230, Pakistan
3   Visual Computing Center, King Abdullah University of Science and Technology (KAUST),
    Thuwal 23955, Saudi Arabia
4   Department of Information Technology, The University of Haripur, Haripur 22620, Pakistan
*   Correspondence: numan.ali@mail.au.edu.pk

**Abstract:** In science education laboratory experimentation has a vital role for students' learning enhancement. Keeping in view the importance of modern day technologies in teaching learning process, various interactive laboratories (ISLs) have been developed to assist students in hands-on experiments in science education. In this paper we describe the potential contributions of existing interactive science laboratories (ISLs) in the major subjects of science, i.e., chemistry, biology and physics. The existing ISLs include virtual labs and simulation software where users performed their experiments. Important problems and challenges in the existing ISLs are highlighted. The systematic literature review (SLR) methodology is used for article searching, selection, and quality assessments. For this study, 86 articles after final selection using SLR are selected and classified into different categories. Each article is selected after briefly studying its different information, including category of the article, key idea, evaluation criterion, and its strengths and weaknesses. A subjective study with field experts was also conducted to investigate one of our existing virtual lab about the practical implementation and to find out the key issues in its implementation and use. Then, considering the suggestions of the subjective study, some guidelines are proposed for the improvement of future ISLs.

**Keywords:** interactive learning; virtual science laboratories; computer simulation-based experiments; adaptive aids; cognitive load

## 1. Introduction

Interactive teaching plays a vital role in the creation of students' conceptual learning due to which interactive teaching has become an integral part of teaching and learning [1]. The advancement of the computer is exposing the education system to new ways, where students take more interest, allow them to use the new tools and to motivate them for learning [2,3]. In this context the use of virtual labs (VLs) and computer simulation-based labs (CSLs) provide state-of-the-art solution for problems in science education, where the physical alternative is not available, doing the actual work is costly or very dangerous.

### 1.1. Purpose and Objective of the Study

The aim of this research is to study the contribution of ISLs in science education in general. Following are the main objectives of this study:

- To study the benefits of interactive labs in science education.
- To study the existing interactive labs in science education.
- To describe problems and challenges of the existing interactive labs.
- To conduct a subjective study by interviewing the field experts.
- To propose solutions for the improvement of interactive labs in science education.

### 1.2. Computer in Education

In teaching-learning process innovative approaches have been developed by using computer technology [4]. Using computer technology immersive and interactive environments are created to facilitate or aid learning [5,6]. Computer-based learning environments are often more effective than traditional teaching tools [7]. These activities enable students to experience phenomena through their own eyes, ears and hands rather than through the eyes of a teacher or textbook writer [8,9]. Winn et al. [10] suggested that interaction is a more important facilitator for learning than immersion for some kinds of task. Therefore, the use of computers has become familiar technology in education for students' learning improvement particularly in practical education.

### 1.3. Analytical Reasoning

In the subjects of chemistry, biology and physics hands-on experiments have been less represented due to the difficulties and constraints in offering laboratory activities. The majority of the educational institutions in developing countries are facing several problems in establishing science laboratories. Some of them are the following:

- Arrangement of expensive chemicals, apparatuses and other models such as microbes, protists, seeds, and cells are costly and expensive [2,11].
- The inability of an instructor to evaluate the performance and learning ability of every student [12].
- In the absence of an instructor it is difficult for students to perform their lab work [2,13].
- A little mistake in a real laboratory environment may hurt the student or may cause damage to the laboratory [12].
- Consumption of chemicals/apparatuses and breaking of glassware is also an issue [2,13].
- Repetition of instruction to students is also an issue for instructor [14].
- Some time it is difficult to fully explore microscopic objects with precision in real experiments [15].
- Repetition of an experiment will require more time and resources [13].
- The psychological factors such shyness or embarrassment involved in requesting an instructor to repeat an experiment [15].

### 1.4. Computer in Science Education

Computer-based learning is one of the most imperative contrivances that support students learning in different fields of science such as Virtual Reality Physics Simulation (VRPS) [16], Construct 3D for Mathematics Education [17], Virtual ChemLab Projects [18] and biological education [19] etc. So far, one of the solutions of the above problems and limitations is the use of computer technology in science education. Hands-on experiments of science subjects are among the difficult tasks to be performed by students in laboratories [20,21]. Interactive laboratories (ISLs) have been used efficiently as alternative or preliminary activities for hands-on experiments in high school and college level science courses, respectively. The students' performance and learning capabilities can be enhanced by using virtual and simulation-based experiments [22,23]. Therefore, the use of virtual labs and simulation-based experiments have become familiar technology in science education for students learning enhancement.

### 1.5. Virtual Science Laboratories

Virtual science labs (VSLs) are computer-created environments resembling a real science laboratory or room (i.e., virtual chemistry lab, virtual biological lab and virtual physics lab) in which users can move from one position to another, they can view the environment from around, touch models/skeletons, chemical and glassware (i.e., test tubes, thermometer and voltmeter etc.) and manipulate the models and glass wares [24]. In VSLs users can select and manipulate objects and chemicals and can also simulate their experiments by real time interaction [25,26]. VSLs are used for both low and high level laboratory activities,

as an alternative or pre-requisite to physical laboratory [20,27]. In distance learning education VSLs are mostly used for the simulation of experiments, because it can be accessed from any place [22,27]. In VSLs students can perform their lab works repeatedly without any health risks and costs [28]. VSLs are also very suitable for the exploration of very small (microscopic information), big or dangerous chemicals that cannot be explored in the normal situation of the physical world [19,29]. For instance, to perform a particular experiment in an interactive science lab which all the essential models/skeletons and glass wares (slides, test-tubes, pipettes, retorts etc.) are distributed in their corresponding placements (shelves, tables, tube box etc.). Users select the apparatus and glassware from the corresponding positions which are required for the selected experiment. Users can interactively simulate their experiments according to the correct procedure. In VSLs, users can also quantify the required measurements by means of virtual measuring apparatus such as pipette and test tubes.

*1.6. Computer Simulation-Based Experiments*

Computer simulation-based experiments (CSEs) are also a form of interactive learning interface in which the operation of a physical-world experimental process are imitated on a computer [30,31]. In a physical-world if it is dangerous to investigate the behavior of an object or system CSEs are used as alternative tools in which investigations are carried out virtually. They use the mathematical description or model to find analytical solutions to problems which enable the prediction of the behavior of a real system from a set of parameters and initial conditions [32]. CSEs are used in a wide variety of practical contexts, such as biology, chemistry and physics [33,34]. For instance, to investigate nuclear blasting to find out its various elements represented with a mathematical description or model that takes into consideration various elements such as heat, velocity, and radioactive emissions.

*1.7. General Benefits of Interactive Labs*

The general benefits of ISLs are the following:

- The main benefit of using ISLs to provide safe and control environments, where users can perform a task without any risk and hesitation [21].
- ISLs provide the sensation of a real world's environments where users can interact with objects in real time [24,35].
- ISLs can be accessed from remote places thus allowing collaborative work .
- ISLs can be used for the visualization of small and complex problems such as to study the structure of a molecule, atom or biological cell [33].
- ISLs are very suitable in those cases where the actual execution of a work is dangerous to perform, for example, the simulation of acid-based experiments [34,36].
- ISLs also provide different learning styles which make learning fun and interesting [20].
- ISLs train many people at one time [37].

*1.8. Issues of Interactive Labs*

There are also some potential drawbacks and problems arising with virtual environments.

- ISLs are costly to implement in educational organizations because they require specialized sensory-motor interfaces [38].
- It is quite difficult for users to perform a computer-based learning task on their desk as they do in traditional classrooms. Therefore, ISLs also require proper personal spaces for users to comfortably complete their learning using ISLs [39].
- Experience in ISLs may generate carelessness, lack of seriousness and irresponsible attitude in students [40].
- It is a digital experience in ISLs that gives a real which nobody can see because it doesn't exist in the real world. Therefore, it is a fact that the last stage in training usually requires real environment and equipment and it is the only way to obtain actual skills through hands-on practice [41].

*1.9. Related Surveys*

The systematic literature review (SLR) methodology [42] is a predefined series of steps, used for review conducted in the field of medical and social sciences, and software engineering. There are few survey papers in related topics, however, there is no SLR on ISLs for chemistry, physics and biology experiments. The only one SLR [43] we found is about gamificiation in science has only 24 article after final selection.

Bellou et al. [44] reviewed empirical research on digital learning technologies and their applications in primary and secondary chemistry education, during the period 2002 to 2016. They emphasized the pedagogical value of digital learning technologies in chemistry education. However, this study is specific to chemistry education. Similarly, there have been some other surveys covering only a specific domain, such as a comparative review of the existing virtual chemistry laboratories (VCLs) by identifying their strengths and shortcomings and formulate guidelines for the development of future VCLs [14], a review on augmented reality (AR) to identify the real situation of AR developments and its potential for three-dimensional (3D) visualization of molecules [45], a review on virtual reality (VR) in the visualization of atomic/molecular [46]. Other surveys such as immersive VR in physics education [47], analysis of VR in physics learning [48], and literature review on the effect of 3D display technologies in biology education [49]. Similarly, Byukusenge et al. [50] conducted a comprehensive literature review on the influence of virtual labs in biology education. However, this study is related to specific topics in biology education i.e., cell and genetics.

Our current study is with a different methodology and narrow direction covering a total of 86 articles after final selection. Similarly, we have a cross validation of the SLR findings using a subjective study (i.e., experts feedback). In addition, we also focused on the analysis of existing ISLs using certain factors, including educational and technological context and content, feasibility, challenges, and their working mechanism. Particularly, our method has the following distinctions over previous surveys:

- We conducted an SLR that covers all the existing literature on existing ISLs.
- To avoid any possible research bias, we followed a well-defined method for searching of articles, inclusion/exclusion, quality assessment, and data extraction.
- We analyzed problems and challenges in the existing ISLs.
- We conducted a subjective study with field experts to investigate about the practical implementation of our existing virtual chemistry lab [20] and to find out the key issues in its implementation and use.
- Considering the suggestions of the subjective study, we also suggested novel solutions for the forthcoming ISLs.

The rest of the paper is organized as follows. This section (i.e., Section 1 describes purpose and objective, background and importance etc. of our current study. Section 2 describes research methodology of the study, considering our research questions and the extracted data etc. Section 3 elaborates related studies about ISLs in the field of science subjects. Section 4 declares constraints in the existing science laboratories. Section 5 describes the feedback and suggestions from field experts and the proposed solutions for the improvement of future interactive science laboratories. Section 6 provides discussion and few interesting proposed solutions. Finally Section 7 is related with conclusion and limitation.

## 2. Research Methodology

Inspired by the popularity of SLR, for this research work, we used SLR methodology [42] during study the literature related to the use of ISLs in science education (i.e., chemistry, biology and physics). It is a systematic approach which is used for identification, evaluation, and interpretation of the related published articles with their references and particular research questions. In SLR, the rules are strictly followed during whole process of the review [42,51] (i.e., searching criteria, analyses, quality assessment and selection of the articles). SLR methodology is mostly followed in the field of software engineering, how-

ever, with passage of time it is being implemented in other fields as well [51,52]. Therefore, for this research study, the literature related to the use of ISLs in science education (i.e., chemistry, biology and physics) published as full length articles both in scientific journals and conferences during 2000 to 2022 were studied. Figure 1 illustrates the overall pipeline of our study which is further described in the following subsections.

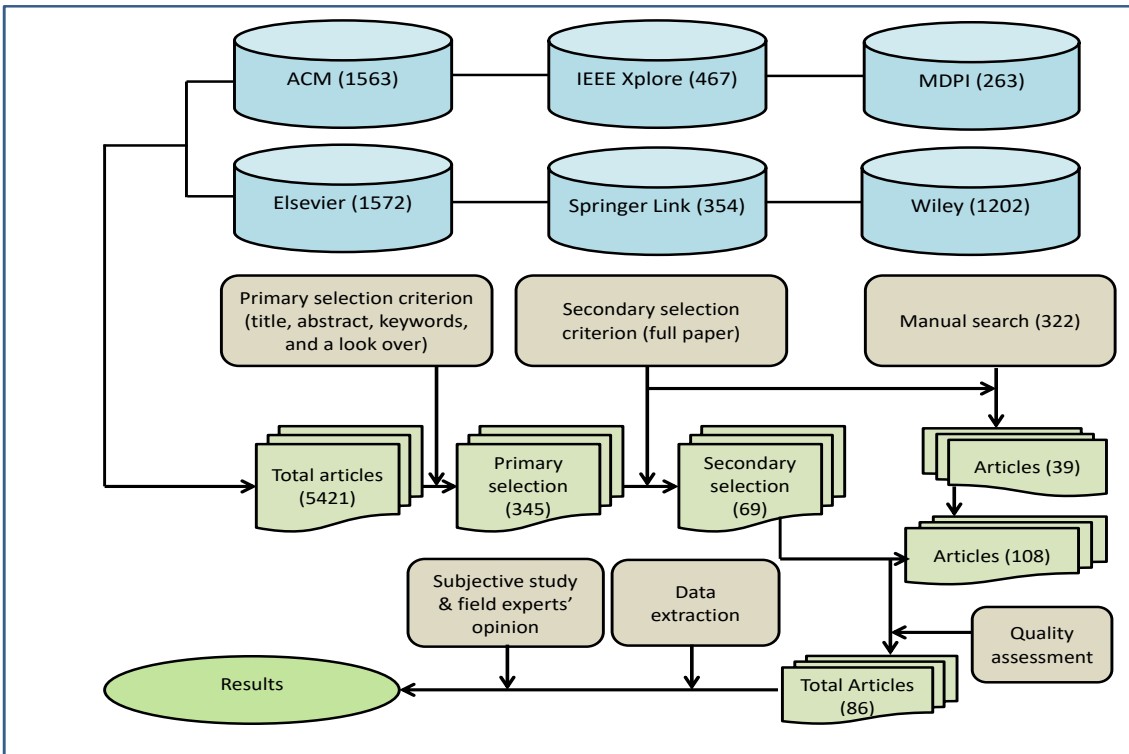

**Figure 1.** Research methodology, including search strategy, article selection criteria, quality assessment, and data extraction.

### 2.1. Research Questions

We conducted an analysis of existing ISLs in major science subjects using some factors such as technological context and content in science education, feasibility, issues and working mechanism. Particularly, our research questions are the following:

- RQ 1: What are the drawbacks of experiments in physical science laboratories?
- RQ 2: What are the potential contributions of ISLs in science education?
- RQ 3: What are the issues and constraints in existing ISLs?
- RQ 4: What are the assessments of field experts by subjective study?
- RQ 5: How to improve the forthcoming ISLs by interviewing the field experts?

### 2.2. Search Strategy for Articles

Search strategy plays an important role to collect the relevant literature on a specific topic in an efficient manner. The relevant articles were found by searching various digital research forum databases, including ACM, Elsevier, IEEE, MDPI, Springer Link, and Taylor & Francis. Moreover, reputed journals regarding education such as the Journal of Chemical Education, Journal of Universal Computer Science and Computers & Education published by non-profit organizations were also searched as well. In the first part of searching, we used the following combination of keywords and their synonyms using Boolean operators. We refined and finalized the following search string.

- For interactive chemistry labs (ICLs) such keywords are used: "chemistry" AND "education" AND "computer" OR "chemistry" AND "education" AND "virtual chemistry labs" OR "chemistry" AND "experiments" AND "simulations".
- For interactive biology labs (IBLs) such keywords are used: "biology" AND "education" AND "computer" OR "biology" AND "education" AND "virtual biology labs" OR "biology" AND "experiments" AND "simulations".
- For interactive physics labs (IPLs) such keywords are used: "physics" AND "education" AND "computer" OR "physics" AND "education" AND "virtual physics labs" OR "physics" AND "experiments" AND "simulations".

We used six different libraries (as shown in Figure 1) for articles search using above string.

### 2.3. Inclusion and Exclusion Criteria

As a second part, we searched for articles cited in the papers we read and were followed two phases such as primary and secondary selection phases for the article selection criteria. In the primary selection phase, we read the title, abstract and keywords of each article and included relevant articles. In the secondary selection phase, we read all articles and excluded articles based on the following inclusion and exclusion criteria:

- The articles related to interactive science labs, specifically ICLs, IBLs and IPLs, were included and those articles which were not related to interactive science labs were excluded.
- The articles which follow an explicit research methodology and present empirical results were included and the articles with no empirical results were excluded.
- The articles in peer-reviewed journals and conferences were included and non-peer-reviewed reports and books were excluded.
- Articles written in English were included and articles presented in other languages or non-English language articles were excluded.
- Duplicated articles (most of materials matching with other articles) were excluded.
- Theoretical proposals or opinion articles were excluded.
- We also conducted a quality assessment (see next Section 2.4 and articles with assessment scores less than 2 were excluded.

Finally, 86 studies referring to ICLs, IBLs and IPLs were included in the literature review. The literature review is divided into the following six categories and classification of the existing ISLs is also shown in Figure 2.

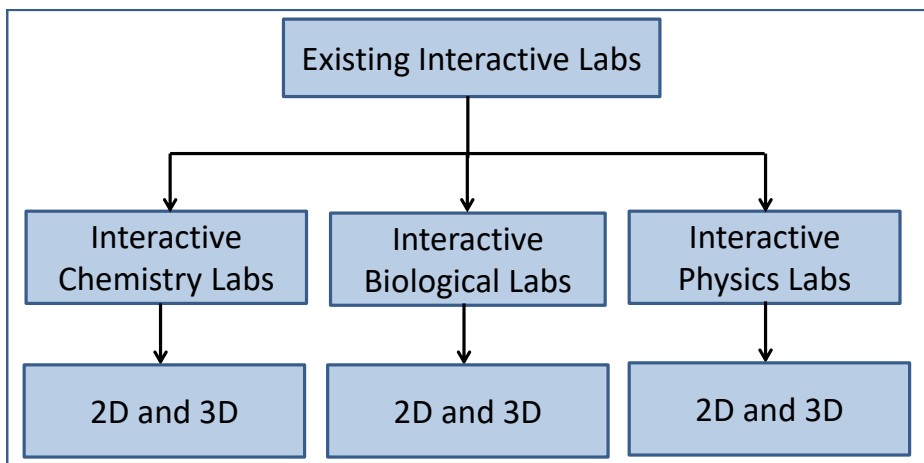

**Figure 2.** Classification of the existing ISLs.

### 2.4. Quality Assessments of Articles

Each article was analyzed based on its quality, contents and venue of publication. Similar criteria were also used in a recent SLR article [51]. We calculated the article quality by numeric values i.e., from 1 to 4, with answering the following four questions.

- Q1: How many citations of the article per year using Google Scholar?
- Q2: Is publication of the article in a standard journal/conference?
- Q3: Is the method in ISLs innovative, helpful and relevant to the community?
- Q4: Are the user study and results examined appropriately (i.e., an unbiased user study)?

The answer for each question was selected on a rating scale of the three different options, i.e., "good", "average" or "poor". The numerical values of all answers were added to calculate the accumulative assessment score for each article. The accumulative assessment score for each article was calculated by adding the numerical value for "good", "average", or "poor" options were "1", "0.5" and "0", respectively. Articles with 2 or more accumulative scores were included and the remaining were excluded. Articles with per-year citations $\geq 3$ were rated as "0.5" and the remaining were rated as 0 for question 1. For question 2, articles from standard and relevant journals/conferences were rated as "1" and the remaining articles from multidisciplinary lower reputation venues were rated as "0.5" or "0". Similarly, questions 3 and 4 were also answered after thoroughly reading the articles based on their contributions, novelties, fairness and completeness of the results. In the quality assessments phase 22 articles were excluded.

*2.5. Data Extraction*

We extracted the following information from each selected article based on our research questions:

- The proposed solution basis on the key issues in the selected articles.
- Interaction devices for simulation of experiments in the exiting ISLs.
- Guidance (assistance aids) during simulation of experiments in the exiting ISLs.
- The pros and cons of the existing ISLs and possible future directions.

We analyzed the collected data for data extraction, and considered the nature of interactive science labs in each articles as a separate category. In this manner, the articles are classified into six different categories which are presented in next section (i.e., Section 3).

## 3. Existing Interactive Labs in Science Education

This section presents existing ISLs both in virtual and simulation-based experiments in science education i.e., chemistry, biology and physics including two-dimensional (2D) and three-dimensional (3D)-based laboratories.

*3.1. Interactive Chemistry Laboratories*

This subsection presents Interactive Chemistry Laboratories (ICLs) both in 2D and 3D environments.

3.1.1. Interactive Chemistry Laboratories in 2D

In 2000, The Virtual Analytical System (VAS) is a 2D lab to train the students about the usage of spectrometer in a lab work. Students can improve their practical skill about the operation of spectrometer in a proper way during their laboratory works. The system is suitable only for the operation of a spectrometer [11]. In 2003, an online-tutorial-based 2D VCL developed by Climent-Bellido et al. [53]. In this system students can improve their laboratory skills for real chemistry experiments. The system is useful to achieve essential information such as operation and other properties of chemicals and apparatus in lab works and to motivate the students for habitual lab activities. A 2D virtual laboratory called Virtual Unit Operational Laboratory (VUOL) has developed for the operation of various industrial equipment. Students can learn the controlling and operating skills of different industrial equipment such as double-pipe heat exchanger method, gas absorber method, and a cooling tower method [54]. For collaborative chemistry experiments a virtual lab, which is known as VLab, is used for collaborative tasks in a lab. Through VLab students use their separate computers to collaboratively select chemicals and glassware for the selected experiment. They can also use a simple chat box for collaboration among them [55].

Cengiz [56] developed the 2D Virtual Chemistry Lab (2D VCL) for high school chemistry experiments to deal the issue of lack of chemicals and equipment in physical laboratories. Through this experimental study the author found that virtual labs had positive role to overcome the issue lacks in physical labs. The ChemCollective [57] is a web-based platform having multiple virtual labs. Users can learn through various sources such as 2D tutorials, scenario-based learning activities, and concept tests [58]. A study has been conducted by Akaygun [59] to compare static and dynamic representations of models of oxygen atom. In this study three animation-developing software were used i.e., ChemSense, K-Sketch and pencil for drawing oxygen atomic model and motion of electrons in orbitals around nucleus. Evaluation revealed that animation softwares are important tools in science classes and could help science educators in making of students' mental models.

In 2018, Ryoo et al. [60] developed interactive molecular visualizations system for high school chemistry. In this system an instructor guides the students through questions posed and interpreted by the students to explicitly illustrate the molecular processes of chemical phenomena. Similarly, Aljuhani et al. [37] developed a web-based 2D platform where authors have claimed the simulation of different experiments of middle school level chemistry but have not provided explicit details. Edraw [61] has developed a web-based 2D Vector Chemistry Laboratory (2D VCL) for symbolic representation of chemical equations, structures, and molecular formulas through various examples and templates. It is also suitable for the sketching of chemical map. In 2D VCL the interaction is carried out via 2D graphical interfaces [62]. Similarly, a web-based 2D VCL developed by Softpedia for higher secondary school chemistry experiments. It is also used for the simulation of chemical reactions in which users select a container and the required substances for the selected experiment. It also provides the periodic table, with valuable information about chemical elements [63,64]. Using an online 2D interface (i.e., PhET) developed by University of Colorado Boulder for high school and university level chemistry education. Students can learn the method of measuring and calculating the pH or electrode conductivity of an acid or base. The system is suitable for knowing the similarities and differences between strong acids and weak acids or strong bases and weak bases [65]. In Model ChemLab students can simulate the concept of chemical reactions. In this system students can also learn the operations of different chemicals and apparatuses. ChemLab [66] is a 2D interface in which users carried out the selection of experiments, its chemicals, and equipment and the required quantity of chemicals through menus and dialogue boxes [67]. The analysis of the existing 2D-ICLs is summarized in Table 1.

**Table 1.** Analysis of the existing Interactive Chemistry Laboratories in 2D.

| S. No | Reference | Year | Interaction Devices | Guidance (Assistance) | Venue | Remarks |
|---|---|---|---|---|---|---|
| 1 | Waller et al. [11] | 2000 | Mouse + Keyboard | None | Journal | In this system students can improve their practical skill about the operation of spectrometer and its use in laboratory works. |
| 2 | Climent-Bellido et al. [53] | 2003 | Mouse + Keyboard | None | Journal | Students can improve their laboratory skills for actual experiments using tutorials-based information about essential glass wares and other equipment. |
| 3 | Vaidyanath et al. [54] | 2007 | Mouse + Keyboard | None | Journal | The system is used that how to control and operate the different industrial equipment such as double-pipe heat exchanger method, gas absorber etc. |

**Table 1.** *Cont.*

| S. No | Reference | Year | Interaction Devices | Guidance (Assistance) | Venue | Remarks |
|---|---|---|---|---|---|---|
| 4 | Cengiz [56] | 2010 | Mouse + Keyboard | None | Journal | It is suitable for the simulation of high school chemistry experiments and to improve students' practical skills. |
| 5 | Tsovaltzi et al. [55] | 2010 | Mouse + Keyboard | Text-based guidance using chat box | Journal | In this system users can collaboratively simulate their chemistry experiments remotely and can cooperate with one another using chat box to improve their practical skills. |
| 6 | Yaron et al. [58] | 2010 | Mouse + Keyboard | None | Journal | In this system users can learn through various sources such as 2D tutorials, scenario-based learning activities, and concept tests. |
| 7 | Akaygun [59] | 2018 | Mouse + Keyboard | None | Journal | In this study three animation-developing software were compared i.e., ChemSense, K-Sketch and pencil for drawing oxygen atomic model and motion of electrons in orbital around nucleus. |
| 8 | Ryoo et al. [60] | 2018 | Mouse + Keyboard | Inquiry-based instruction | Journal | The system is feasible only for molecular visualizations to explicitly illustrate the molecular processes of chemical bonding. |
| 9 | Bazurin [62] | 2020 | Mouse + Keyboard | None | Journal | The system is used for symbolic representation of chemical equations, structures, and molecular formulas through various examples and templates. |
| 10 | Ali et al. [64] | 2021 | Mouse + Keyboard | None | Journal | In this system students can learn the concept of quantity of chemical and its reactions. |
| 11 | Taibu et al. [65] | 2021 | Mouse + Keyboard | None | Journal | The system is very suitable to learn the method of measuring and calculating the pH or electrode conductivity of an acid or base. |
| 12 | Aljuhani et al. [37] | 2018 | Mouse + Keyboard | Text-based guidance | Journal | This system is used for the simulation of different experiments on middle school level chemistry. |
| 13 | Hernández-Garces et al. [67] | 2021 | Mouse + Keyboard | None | Journal | In this system students can learn the operations of different chemicals and apparatuses to improve their skills for actual chemistry experiments. |

### 3.1.2. Interactive Chemistry Laboratories in 3D

CSU ChemLab is a virtual lab in which students can learn the assembly of various equipment (apparatuses and glassware) required in a experiment. It is a suitable VCL to learn only the procedure of experiments, but the simulation of chemical reactions is not possible in it which is the limitation of CSU ChemLab [68]. In 2004, Girault et al. [69] developed an online VCL in which students can remotely enter the required data for experiments and then a tele-robot performs the experiments according to the provided data. Similarly, the Virtual Reality Undergraduate Projects Laboratory (VRUPL) is a 3D VCL in which undergraduate students are trained for physical experiments. In VRUPL students can learn about apparatuses and glass wares and their proper assembly in a particular experiments. Moreover, students are also guided about the safety rules about habitual chemistry experiments, both in industrial and educational contexts. The system is very helpful for learning safety rules, but it does not provide the simulation of chemical reactions [70]. Woodfield et al. [71] conducted a study and found that Virtual ChemLab

had positive effects on the performance of students and also improve the problem solving capabilities in them. LabVIEW is a virtual lab in which students can learn that how to operate an isoteniscope and learn the method of measuring the vapor pressure of a liquid in it [72].

In 2007, Limniou et al. [73] developed a fully immersive chemistry environment for learning the procedure of chemicals reaction of molecules. In this system an expensive hardware called Cave Automatic Virtual Environment (CAVE) is used for fully immersion. In this system users can react chemicals in simulation form through which they can visualize the 3D model of molecules. Similarly, Stone has developed a VCL for chromatography in which users can learn the technique for the separation of a mixture. In this system users can improve their skill in chromatographic techniques by using gas chromatography (GC) and high-performance liquid chromatography (HPLC) simulators. The author reported that his system is very suitable for students in chemistry education to improve their skills in chromatographic-based experiments [74]. In Virtual Reality Interactive Learning Environment (ViRILE) students can learn about the operation of chemical plant and its various components. Students can also learn the experimental procedure that how to simulate the process of chemicals reaction in chemical plant. The ViRILE is better system to improve students' practical skills for actual experiments using chemical plant [75].

Eman et al. [76] developed an online collaborative 3D virtual class room that consists of a virtual periodic table in which users can interact collaboratively where they can also use humanoid avatars for collaborative communication and audio-based aids. Students can explore different visual information of chemical elements such as physical and chemical properties and 3D visualization of atomic structure. Furthermore, iVirtualWorld is an online VCL with a graphical user interface (GUI) allowing users to set different properties of the required chemicals and glass wares for the selected experiment, which makes the interface difficult for users to perform the experiments [3]. The Valence Shell Electron Pair Repulsion (VSEPR) is an online virtual chemistry classroom developed by Kenney and Merchant [77] for 3D rendering of molecules and ions. The system is used for studying the 3D structures of molecules and ions which is very helpful for students' learning enhancement. In 2014, a study conducted by Winkelmann et al. [78,79] to compare the performance of students in their proposed VCL called Second Life (SL) and habitual laboratory. They found that there is no significant difference in students' performance in both environments, however, in the virtual environment students took less time in completion of chemistry experiments. The evaluation also revealed that VCL is very useful for motivation of students and to make them self-sufficient in performing of experiments. For high school chemistry experiments a 3D interactive VCL was proposed by Ali et al. [80] where user can interact with the system via a haptic device i.e., Nintendo wiimote. Evaluation revealed that 3D interaction can improve students' motivation and practical skills for chemistry experiments. In addition, it also provides multi-modal information both in visual and audio forms about chemical elements [81]. Jagodziński and Wolski [82] have developed a 3D VCL that provides gestural interaction (i.e., Natural User Interface) and conventional video-based instructions. They found that in 3D VCL both the NUI and conventional video-based instructions had positive effectiveness on improving the sense of self-efficacy in students. In a collaborative virtual learning laboratory (CVLL) students simulate chemistry experiments by a task distribution module. In CVLL an experimental task is distributed on participants to perform the whole experiment in a collaborative mode to improve their learning capabilities and performance [83]. Multi-modal virtual chemistry laboratory (MMVCL) has investigated the effectiveness of procedural guidance (i.e., textual instructions) on students' performance during simulation of experiments. The authors found that procedural guidance is more effective on students' performance through which they can complete their experiments without any instructors/teachers [20]. To evaluate the effect of fuzzy logic approach in 3D-VLEs, Alam et al. [84,85] developed a fuzzy-logic-based virtual environment. LateNite Lab (LNL) is another online VCL [86] where secondary school students can simulate their experiments via 2D graphical interactive interfaces [87].

In 2019, Wu et al. [88] developed 3D virtual chemistry lab for simulation of titration-based experiments. The authors reported that their proposed virtual reality chemistry lab is suitable to promote users' learning confidence. Sustainable innovation experiential learning model (SIL) has been proposed by Su and Cheng [89] to investigate students' experiential learning, mental loading, and self-efficacy in virtual chemistry experiments. The authors found that extra chemical equipment which are not used in the current experiment make the environment complex and puts more mental load on students that affects students' performance during simulation of chemistry experiments. In 2022, Ali et al. [90] developed Purpose-built Virtual Chemistry Lab (PbVCL) with arrow-textual aids. PbVCL displays only the specific chemicals and glass wares, used in the current experiment while hiding other equipment to minimize the cognitive load. The arrow-textual aids assist the users/students to simulate an experiment in a VCL correctly according to the procedure. The analysis of the existing 3D-ICLs are summarized in Table 2.

**Table 2.** Analysis of the existing interactive chemistry laboratories in 3D.

| S. No | Reference | Year | Interaction Devices | Guidance (Assistance) | Venue | Remarks |
|---|---|---|---|---|---|---|
| 1 | Barney et al. [68] | 2003 | Mouse + Keyboard | None | Symposium | It is a suitable VCL to learn only the procedure of experiments, but the simulation of chemical reactions is not possible in it. |
| 2 | Girault et al. [69] | 2004 | Mouse + Keyboard | None | Journal | Through this system students can remotely enter the required data for experiments and then a tele-robot performs the experiments according to the provided data. |
| 3 | Bell [70] | 2004 | Mouse + Keyboard | None | Conference | Through this system students can learn the safety rules about physical chemistry experiments. |
| 4 | Woodfield et al. [71] | 2005 | Mouse + Keyboard | None | Journal | Through this system the authors found that VILs had positive effects on the performance of students and also improve the problem solving capabilities in them. |
| 5 | Belletti et al. [72] | 2006 | Mouse + Keyboard | None | Journal | This system is useful only in teaching how to operate an isoteniscope and to learn the method of measuring the vapor pressure of a liquid in it. |
| 6 | Limniou et al. [73] | 2007 | Flystick | None | Journal | In this CAVE-based system users can learn the concept of chemicals reaction and the visualization of molecules in 3D form. However, CAVE is more expensive technology. |
| 7 | Stone [74] | 2007 | Mouse + Keyboard | None | Journal | The system is only suitable for improving students' skill about chromatographic-based activities in a lab. |
| 8 | Schofield et al. [75] | 2010 | Joystick | None | Journal | In this system students can learn about the operation of a chemical plant and its various components. |
| 9 | Eman et al. [76] | 2012 | Mouse + Keyboard | Collaborative verbal guidance | Conference | This system provides only the visual information of chemical elements including chemicals properties and atomic structure in 3D visualization but it does not provide simulation of experiments. |

**Table 2.** *Cont.*

| S. No | Reference | Year | Interaction Devices | Guidance (Assistance) | Venue | Remarks |
|---|---|---|---|---|---|---|
| 10 | Zhong et al. [3] | 2013 | Mouse + Keyboard | None | Journal | In this system users can perform chemistry experiments using menus such as 2D graphical user interface (GUI). |
| 11 | Keeney and Merchant [77] | 2013 | Mouse + Keyboard | None | Symposium | The system is used for studying the 3D structures of molecules and ions but it does not provide simulation of experiments. |
| 12 | Winkelmann et al. [78] | 2014 | Mouse + Keyboard | None | Journal | Through this system authors have found that VILs are better interfaces for enhancement of students' practical skill and motivation. |
| 13 | Ali et al. [80] | 2014 | Wiimote | None | Conference | This system provides 3D interaction that can improve students' motivation and practical skills for chemistry experiments. |
| 14 | Jagodzinski and Wolski [82] | 2015 | Kinect XBOX | Video-based guidance | Journal | Through this system authors studied the effectiveness of using hand's gestures in 3D VCL and found that NUI enhances the sense of self-efficacy in students. |
| 15 | Khalid et al. [83] | 2016 | Wiimote | None | Journal | In this system students can simulate chemistry experiments collaboratively by a task distribution module. |
| 16 | Ullah et al. [20] | 2016 | Wiimote | Procedural guidance | Journal | In this system students can simulate their experiments by procedural guidance. |
| 17 | Alam et al. [84] | 2017 | Mouse + Keyboard | None | Journal | This system is fuzzy logic-based 3D-VLEs that is very suitable only for theoretical chemistry but it does not provide simulation of experiments. |
| 18 | Faulconer and Gruss [87] | 2018 | Mouse + Keyboard | None | Journal | In this system students can simulate their high school chemistry experiments via 2D graphical interactive interfaces. |
| 19 | Wu et al. [88] | 2019 | Leap Motion | Textual Tips-based Guidance | Journal | The system is suitable only for the simulation of titration-based experiments by using hand gestures. |
| 20 | Su and Cheng [89] | 2019 | Mouse + Keyboard | Textual-based Guidance | Journal | It is used for high school level chemistry experiments and guides students by textual-based instructions during performing experiments. |
| 21 | Ali et al. [90] | 2022 | Mouse + Keyboard | Arrow-Textual Aids | Journal | In PbVCL displays only the specific chemicals and glass wares, used in the current experiment while hiding other equipment to minimize the cognitive load. |

### 3.2. Interactive Biological Laboratories

This subsection presents Interactive Biological Laboratories (IBLs) both in 2D and 3D environments.

### 3.2.1. Interactive Biological Laboratories in 2D

Virtual Biology Experiments (ViBE) is a 2D virtual biology lab for university level biological lab works that assists students by textual guidance about various biological experiments such as to study cell features, whole organisms, separation of cellular com-

ponents and quantification of cell division and measurement of enzyme activities. It also assists students to improve their technical skills about some biological lab instruments such as centrifuges and microscopes. ViBE is a useful virtual lab for the bio experiments. On one side the lab reduces the instructor's involvement and on the other side students interest and adaptability to various activities is enhanced [91].

In 2004, Evans et al. conducted a study on the efficiency of Virtual Lecture (VL) in the field of biological sciences. In this study two types of learning materials were compared i.e., the VL teaching materials compared with web-pages-based teaching materials regarding genetics and reproduction topics in biology. The analysis revealed that VL-based teaching materials are powerful and flexible learning tools as compared to web-page-based teaching materials [92]. A 2D game called CellCraft has been developed by Peter and Pecore [93] for primary level students that introduces organelles of the cell in a stepwise manner. It is a helpful interactive online tool for students to read and brief them about the cell fend off viruses, damage from free radicals, and other dangers, through an increasingly complex set of tasks. In 2012, Muhammad et al. [94] developed VLab-Bio which provides visual-based aids (including images, audio and videos) regarding cell division. The VLab-Bio is used for the experiments of cell division including chemical composition of cell, structure of cell and cell cycle etc. The evaluation revealed that VLab-Bio is a supporting tool in teaching and learning of biology for both teachers and students. Virtual Urchin [95] is an online framework, that provides interactive tutorials about some topics of biology such as microscope, specimen comparison, analyzing gen function, acidifying ocean and organisms of sea urchin predators and prey [96]. Biology Labs Online [97] is an online 2D lab which consists of 12 inquiry-based interactive topics which covers standard laboratory topics in introductory biology for both college and university level students. The Biology Labs Online is used by hundreds of colleges and universities where students prepare for further hands-on biological experiments [98]. Similarly, iNeuron is a collaborative iOS mobile app that provides to students the basics of neuroscience through a combination of scaffolded instructions that supports content knowledge and inquiry-based challenges in which students build neural circuits to solve problems collaboratively [99]. The iNeuron is a robust game-like learning app where students connect their devices through Bluetooth or Wi-Fi to solve the problems collaboratively. It is very useful to supplement classroom instruction for neuroscience in biological education. Similarly, android-based virtual biology laboratory is developed (VLab-Bio) based on Biology, Technology, Engineering, and Mathematics (BTEM) for understanding the concept of bacteria. The system is called by the name of BTEM-Based VLab-Bio where high school level students can perform their biology lab work for observing the making of bacterial culture media, bacteria, counting bacterial colonies, and staining gram-positive and gram-negative bacteria [100]. The analysis of the existing 2D-IBLs is summarized in Table 3.

**Table 3.** Analysis of the existing interactive biological laboratories in 2D.

| S. No | Reference | Year | Interaction Devices | Guidance (Assistance) | Venue | Remarks |
|---|---|---|---|---|---|---|
| 1 | Subramanian and Marsic [91] | 2001 | Mouse + Keyboard | Textual Guidance | Conference | It is used for various biological experiments such as study of cell features, whole organisms, separation of cellular components and quantification of cell division and measurement of enzyme activities, centrifuges and microscopes. |
| 2 | Evans et al. [92] | 2004 | Mouse + Keyboard | Lecture Notes | Journal | It offers interactive virtual lecture notes about biological sciences. |
| 3 | Peter and Pecore [101] | 2009 | Mouse + Keyboard | Steps wise instructions | Journal | It is used on primary school level students to teach about the organelles of the cell. |

**Table 3.** *Cont.*

| S. No | Reference | Year | Interaction Devices | Guidance (Assistance) | Venue | Remarks |
|---|---|---|---|---|---|---|
| 4 | Muhammad et al. [94] | 2012 | Mouse + Keyboard | Images, audio and videos aids | Conference | It is used for the school level biological experiments i.e., cell division including chemical composition of cell, structure of cell and cell cycle etc. |
| 5 | Haverkort-Yeh et al. [96] | 2013 | Mouse + Keyboard | Tutorial-based Instructions | Journal | It is used for biology topics such as microscope, specimen compare, analyzing gen function, acidifying ocean and organisms of sea urchin predators and prey. |
| 6 | Son [98] | 2016 | Mouse + Keyboard | Tutorial-based instructions | Journal | It is used only for 12 inquiry-based interactive biological topics both on college and undergraduate level students. |
| 7 | Schleisman et al. [99] | 2018 | Android-based Interface | Collaborative verbal-based guidance | Journal | It is used for the basic knowledge of neuroscience on school level biological education. |
| 8 | Aripin and Suryaningsih [100] | 2020 | Android-based Interface | Textual Instructions | Journal | In this system students can perform their biology lab work for observing the making of bacterial culture media, bacteria, counting bacterial colonies, and staining gram-positive and gram-negative bacteria. |

### 3.2.2. Interactive Biological Laboratories in 3D

In 2002, Friedl [102] designed a VR heart model. Here students can rotate the heart model, move it and zoom in for details. It can be explored during the cardiac cycle to help understand its function. A transparency mode demonstrates the coronary arteries, movement of the heart valves, and blood-flow. In 2003, Mikropoulos et al. [103] developed the virtual learning environments for the exploration of plant cell and the concept of photosynthesis. In this system the internal structure of cell can be explored by navigation, observation and manipulation of different objects. In addition, the concept of organelles and its function in cells can also be explored. Similarly, VRBS (Virtual Reality Biology Simulation) allows to study the structure and function of human eye. In VRBS the iris and pupil of the human eye have been visualized [19]. In 2005, virtual simulation of brain was developed to train surgeons before they arrive at the operating theatre, making surgery safer for patients with less risk of complications. The virtual brain allows the surgeon to feel as though they are actually touching the brain while they operate [104]. A study conducted by Maloney [105] to train female high school biology students about an actual dissection. In this the author used his proposed system i.e., virtual fetal pig dissection and found that the proposed system is a viable tool to train the female students for actual fetal pig dissection. In 2008, Silén et al. [106] developed an interactive 3D visualizations tool for medical students in learning anatomy and physiology. It provides cardiovascular and musculoskeletal topics which are concerning with anatomy and physiology courses. It also provides lectures and demonstration regrading cardiovascular and musculoskeletal topics which are very useful for a conceptual understanding of these topics. VR Teaching Hospital (VR-TH) is an immersive virtual learning environment for teaching human anatomy to medical students. In VR-TH students can walk around, explore the view, watch images and videos displayed on TV screen inside the VR-TH, listen to virtual instructor's lectures and interact with different organs of human anatomy and other characters. The VR-TH system is very helpful for the motivation of learners' curiosities and interests [107].

E-Junior is a serious virtual world (SVW) for primary level students to introduce them the basic notions of natural science and ecology. The system is an effective tool to engage

and satisfy students [108]. In 2012, Cheng and Annetta [109] evaluated students' learning outcomes and their learning experiences through playing a Serious Educational Games (SEGs) that consists a 3D virtual brain model. The SEGs provide a series of brain images to explore the structures of the brain and model their functions to allow players/students for virtually viewing and manipulation. They found that students take more interest in learning by using active immersion in the game worlds. In 2013, Tan and Waugh [110] designed an environment that help students to see DNA, proteins, and cellular structures in 3D space. The results of this study recommend the use of technology in the teaching and learning of molecular biology, especially for male students in Singapore. Borchert et al. [111] have introduced a 'fly through' interface for secondary school biology education. They provide a game-based environment which enables students to think and act like a biologist. The goal is to correctly identify seven organelles inside anarche typical plant cell. However, due to complexity of information and no direct access; the new students may feel worried and confused during interaction. Bonser et al. [112] developed a virtual microscopy system for botany education. The system is suitable to learn the operation of microscope and glass slides with high-resolution digital 'virtual slides'. The system is an effective tool for increasing students' practical skills about the microscope in actual botanical lab works. In 2015, Azhar et al. [113] developed an interaction technique called multi-layered hierarchical bounding-box-based technique for school level biological education. The technique uses a step-by-step and hierarchical approach to have information rich, selection/manipulation and exploration of an object and its constituent parts regarding human skeleton. The technique os multi-layered hierarchical bounding box is very useful and suitable for biological educational virtual environments where students can easily gain rich information regarding complex objects or any human organs. An online BioInteractive Virtual Labs (BIVLs) are developed for high school and college level biological experiments [114]. BIVLs provide lectures, videos, animations and a number of experiments such as virtual labs are Lizard Evolution, Bacterial Identification, Neurophysiology, immunology and cardiology. Students are guided by providing textual guidance, where they can easily simulate their biology experiments according to the correct course of action. BIVLs are very feasible for students to gain experience for hands-on experiments [115]. A problem-based virtual laboratory media is used for the simulation of biogeochemical cycle ecosystems-based experiments. The system is very feasible for the improvement of students' process skills in the experiments and solving of problems regarding biogeochemical cycle ecosystems. Biogeochemical cycle ecosystem is the consideration of the biological, geological and chemical aspects of each cycle on earth [116]. In 2019, Miyamoto et al. [117] developed a 3D virtual lab for undergraduate biological lab works. Students can simulate their five essential laboratory techniques using textual step-by-step instruction regarding biology lab works. These five techniques include preparing of a lab bench, weighing of reagent, dissolving of reagent, adjusting of pH meters and micropipetting. This virtual lab is very helpful for the enhancement of students' performance and understanding in conducting essential biological laboratory techniques. In 2020, Paxinou et al. [118] developed a 3D virtual reality biology lab for photonic microscopy experiments on university level and conducted a study to evaluate the virtual lab with conventional didactic practices. In this study three different teaching methods were evaluated, i.e., a conventional group trained by a simple demonstration of a microscopy procedure, a video group trained by video of the microscopy experiment and a virtual reality group trained by virtual reality microscopy. Evaluation revealed that virtual laboratory simulations are very promising tools for students' conceptual understanding in the domain of microscopy. A Three-dimensional (3D) interactive virtual mustard plant (VMP) has been developed by Ali et al. [119] for secondary school level biological education. In VMP students can achieve visual information about different organs of the mustard plant by list-liner interaction technique using a 3D interactive device. The system is very helpful for students to enhance their learning. In 2021, a study conducted by Sypsas et al. [120] to investigate the students' opinion from two different educational settings i.e., secondary school and university level students. In this

study the authors used their developed virtual reality lab, called OnLabs, for the training of photonic microscope. This study found that both secondary school and university students had a similar attitude towards the virtual lab basis on their learning procedure of a science course. The analysis of the existing 3D-IBLs is summarized in Table 4.

**Table 4.** Analysis of the existing Interactive Biological Laboratories in 3D.

| S. No | Reference | Year | Interaction Devices | Guidance (Assistance) | Venue | Remarks |
|---|---|---|---|---|---|---|
| 1 | Friedl [102] | 2002 | Mouse + Keyboard | None | Journal | It provides a heart model to study the coronary arteries, movement of the heart valves, and blood-flow in heart. |
| 2 | Mikropoulos et al. [103] | 2003 | Mouse + Keyboard | None | Journal | It is used for the exploration of plant cells and the process of photosynthesis. |
| 3 | Shim et al. [19] | 2003 | Mouse + Keyboard | None | Journal | It is used to study the structures and functions of the iris and pupil in the human eye. |
| 4 | Yu et al. [104] | 2005 | Mouse + Keyboard | None | Journal | It is suitable for neurologists in their pre-operative stages regarding brain surgery. |
| 5 | Maloney [105] | 2005 | Mouse + Keyboard | None | Journal | It is useful to explore virtual fetal pig dissection as a learning tool for female high school biology students. |
| 6 | Silén et al. [106] | 2008 | Mouse + Keyboard | None | Journal | It is used for the merging of 3D visualization concept in learning process related to the field of anatomy and physiological education. |
| 7 | Huang et al. [107] | 2009 | Mouse + Keyboard | Virtual instructor | Conference | It is helpful for medical students to improve their knowledge regrading human anatomy to explore different organs. |
| 8 | Wrzesien and Raya [108] | 2010 | Paddle | None | Journal | It is an effective tool for primary level students to study about the ecosystem of Posidonia oceanica. |
| 9 | Cheng and Annetta [109] | 2012 | Mouse + Keyboard | Colors-based guidance | Journal | It is useful for middle school level biology education for exploring the structures of the brain and model their functions. |
| 10 | Tan and Waugh [110] | 2013 | Flystick | None | Static | It is used to enhance students' knowledge regarding DNA, proteins and cellular structures in the field of molecular biology. |
| 11 | Borchert et al. [111] | 2013 | Mouse + Keyboard | None | Conference | It is suitable for secondary school level students to study different organelles inside anarche typical plant cells. |
| 12 | Bonser et al. [112] | 2013 | Mouse + Keyboard | None | Journal | It is useful to improve students' knowledge regarding using of microscopes in botanical education. |
| 13 | Azhar et al. [113] | 2015 | AR ToolKit Marker | None | Conference | It is suitable for the basic information regarding the human skeleton on school level biology education. |
| 14 | Romine and Todd [115] | 2017 | Mouse + Keyboard | Textual-based Guidance | Journal | BIVLs are used for school and college level biology topics such as lizard evolution, bacterial identification, neurophysiology, immunology and cardiology. |

**Table 4.** *Cont.*

| S. No | Reference | Year | Interaction Devices | Guidance (Assistance) | Venue | Remarks |
|---|---|---|---|---|---|---|
| 15 | Syahfitri et al. [116] | 2019 | Mouse + Keyboard | Textual-based guidance | Journal | It is used for the simulation of biogeochemical cycle ecosystems-based experiments on high school level biology education. |
| 16 | Miyamoto et al. [117] | 2019 | Mouse + Keyboard | Visual-based guidance | Journal | It provides five techniques, which includes preparing of a lab bench, weighing of reagent, dissolving of reagent, adjusting of pH meters and micropipetting. |
| 17 | Paxinou et al. [118] | 2020 | Mouse + Keyboard | None | Journal | It is very feasible to train the students about microscopy procedure. |
| 18 | Ali et al. [119] | 2020 | Nintendo Wiimote | List-liner-based guidance | Journal | It is suitable for achieving information about different organs of mustard plant on secondary school level biology education. |
| 19 | Sypsas et al. [120] | 2021 | Mouse + Keyboard | None | Conference | OnLabs is suitable to train the students for the operating of photonic microscope in real lab. |

### 3.3. Interactive Physics Laboratories

3.3.1. Interactive Physics Laboratories in 2D

This portion presents Interactive Physics Laboratories in Two-Dimensional (2D-IPLs).

A Physisc Applets (PhysLets) is a virtual system for secondary school level physics experiments. It provides many physics experiments which consists videos, drawings, symbols and mathematical graphics to create student's mental model for actual experiments. It also facilitates the course instructors to assess and evaluate the student by means of the interactive tests [121]. In 2007, a study conducted by Yang and Heh [122] to investigate and compare the impact of their proposed Internet Virtual Physics Laboratory (IVPL) instruction with traditional laboratory instruction in physics education-based on performance of science process skills, and computer attitudes of. They found that the impact of IVPL on tenth class students is significantly better physics academic achievement than that of those who received traditional instruction.

In 2009, Tlaczala et al. [123] developed Virtual Community Collaborating Space for Science Education (VccSSe) for physics instrumentations-based experiments. The VccSSe facilitates users by simulation-based exercises with dynamic models of physics laws such as Charles's law, Boyle–Mariotte's law, Gay-Lussac's law, heat transportation process, and DC and AC electrical circuits. An Interactive Physics Laboratory (IPL) has been developed by Sabatka [124] for high school level physics experiments. Various experimental groups of students performed the experiments in this environment and the results were compared with habitual methods of teaching and found that IPL have positive effects on students learning. IPL is used for many physics experiments such as electrostatics, optics, motion under gravity, and magnetic field of solenoids etc. Real Time Relativity (RTR) is an interactive game-based simulator that is used in the experiments of quantum mechanics. The system is used for quantum theory such as quantum information, quantum optics and the study of ultra-cold quantum gases (Bose–Einstein). Using this system students take more interest and can improve their skill regrading quantum mechanics in an amusement environment [125]. In 2010, Santos et al. [126] developed a visualization-based environment on the name of Technology Enabled Active Learning (TEALsim) for physics experiments. It is very useful for the visualization of electromagnetism to make visible the magnetic field lines, which are not visible in real setting. TEALsim is very helpful to increase student's analytical and conceptual understanding of the nature and dynamics of electromagnetic phenomena. Wang et al. [127] has developed a virtual physics lab on the name of Model-

based Inquiry Virtual Physics Lab (MBI-VPL) for college level students. The system used for the creation of the MBI-VPL pedagogy method to investigate students' inquiry skills. They designed six learning modules, i.e., topic introduction, actual experiment, virtual experiment, team work, actual applications, and model adjustments. MBI-VPL provides three main physics topics: kinematics and dynamics (i.e., free fall, ballistics, and simple pendulum etc), optics (i.e., mirror reflection and image formation of convex lenses) and electricity (i.e., series and parallel connections and to measure resistance). The results showed that MBI-VPL pedagogy method is more effective for the improvement of students' skills as compared to traditional method. JHKSOFT Electricity Lab (JEL) [128] is developed for middle school level physics experiments. JEL is used for the designing of both electric and electronic circuits-based experiments using electricity components where students can design virtually their own circuit. JEL is provided different circuits-based experiments such as series and parallel circuits, to measure resistance by voltammetry method, to measure the voltage between the ends of a circuit, to accurately measure the resistance by Wheatstone bridge, and to control a circuit by electromagnetic relays etc [129].

Arianti et al. [130] developed a virtual physics lab (VPL) for physics teachers to explore the concept of collision detection. VPL is very useful in distance learning education where physics teachers can easily explore the concept of perfectly elastic collisions, partially elastic collisions and totally inelastic collisions. Dry Cell Microscopic Simulation (DCMS) is a virtual physics lab for exploring the working mechanism of dry cells in producing energy. In DCMS students can see the movement of protons and electrons and they can understand the concept of producing energy from dry cells [131]. A Crocodile Physics (CP) simulator is used for the designing and testing of electronic circuits and to analyze the mathematical and graphical results regarding circuits [132]. Students are also facilitated by tutorial-based materials for improving their theoretical skill regarding electronic circuits [133]. Physics Education Technology (PhET) is an interactive simulation that provides circuit construction kit for the designing of electronic circuits [134]. PhET is used for both secondary school and university level electronic circuit-based experiments and students can simulate different circuit-based experiments such as series circuit, parallel circuit, ohm's law, and Kirchoff's law. Students can also analyze the mathematical equations and graphical results [135]. GO-LAB [136] is also an online virtual physics lab where students use resistors, light bulbs, switches, capacitors and coils to build their own electrical circuits. It also facilitates students by data collection to create graphs of the collected data for analysis [137]. The analysis of the existing 2D-IPLs is summarized in Table 5.

**Table 5.** Analysis of the existing interactive physics laboratories in 2D.

| S. No | Reference | Year | Interaction Devices | Guidance (Assistance) | Venue | Remarks |
|---|---|---|---|---|---|---|
| 1 | Malloy [121] | 2001 | Mouse + Keyboard | None | Journal | Various physics experiments are shown using video metaphor which makes the interaction less. |
| 2 | Yang and Heh [122] | 2007 | Mouse + Keyboard | Textual Instructions | Journal | In this study the authors compared their proposed virtual physics lab (IVPL) with traditional physics instructions and found that IVPL is significantly better than traditional physics instructions. |
| 3 | Tlaczala et al. [123] | 2009 | Mouse + Keyboard | None | Conference | It allows the simulation of various laws of physics such as Charles's law, Boyle–Mariotte's law, Gay-Lussac's law, heat transportation process, and DC and AC electrical circuits. |

**Table 5.** *Cont.*

| S. No | Reference | Year | Interaction Devices | Guidance (Assistance) | Venue | Remarks |
|---|---|---|---|---|---|---|
| 4 | Sabatka [124] | 2009 | Mouse + Keyboard | None | Journal | It is used for the simulation of high school level physics experiments such as electro-statics, optics, motion under gravity, magnetic field of solenoids etc. |
| 5 | Savage et al. [125] | 2010 | Mouse + Keyboard | None | Conference | It is used for the physics experiments such as quantum theory such as quantum information, quantum optics, and the study of ultra-cold quantum gases (Bose–Einstein) using Real Time Relativity (RTR) simulation. |
| 6 | Santos et al. [126] | 2010 | Mouse + Keyboard | None | Conference | It is used only for the visualization of electro-magnetic lines. |
| 7 | Wang et al. [127] | 2015 | Mouse + Keyboard | Verbal-based guidance | Journal | It is used for the simulation of college level physics experiments such as kinematics and dynamics, optics and electricity. |
| 8 | Mirçik et al. [129] | 2018 | Mouse + Keyboard | None | Journal | It is used on middle school level physics experiments that simulates both electric and electronic circuit-based experiments such as such as series and parallel-based circuits to measure voltage and resistance. |
| 9 | Arianti et al. [130] | 2021 | Mouse + Keyboard | Textual Instructions | Conference | The system is very useful to understand the concept of perfectly elastic collisions, partially elastic collisions and totally inelastic collisions. |
| 10 | Wibowo et al. [131] | 2021 | Mouse + Keyboard | None | Conference | It is used for promoting the students' concepts about the workings mechanism of the dry cell in producing energy. |
| 11 | Prastowo et al. [133] | 2021 | Mouse + Keyboard | None | Journal | It simulates electric circuits-based experiments on secondary school level physics experiments. It also provides tutorials for the improvement of students' theoretical skill. |
| 12 | Zulkifli et al. [135] | 2022 | Mouse + Keyboard | None | Journal | It simulates electric and electronic circuit-based experiments on both secondary school and undergraduate level physics experiments such as series circuit, parallel circuit, ohm's law, and Kirchoff's law. |
| 13 | Kapici et al. [137] | 2022 | Mouse + Keyboard | None | Journal | It is used on high school level physics experiments to simulates electrical circuits using resistors, light bulbs, switches, capacitors and coils to build their own electrical circuits. |

### 3.3.2. Interactive Physics Laboratories in 3D

This portion presents Three-Dimensional Interactive Physics Laboratories (3D-IPLs).

In 2005, Inoue et al. [138] developed a 3D virtual physics laboratory using haptic device for interaction. The environment is used for "pulley" experiments in physics to acquire understanding about "balance of force", "work" and "motion equations" using the pulley method in physics. The Phantom Omni device is used for interaction in the simulation of experiments. An online 3D virtual physics lab Edison5 allows multi-users to perform physics experiments collaboratively. It provides graphical data analysis about

electricity and electronics circuits-based experiments to display graphs regarding time, energy, and speed. It also facilitates users through simple video tutorials to improve their theoretical skill [139].

In 2013, Karingula et al. [140] developed Virtual Physics Laboratory (VPL) for high school level physics experiments. VPL facilitates students by automated content delivery using text-to-speech technology to deliver its content in sync with 2D and 3D animations, interactive 3D experiments, and virtual physics instructor for teaching, training and assistance of students. VPL is used for numerous physics experiments such as spring mass system, gravity drop, pendulums, two-masses pulley, and vibration modes of string etc. VPL is a viable tool for delivering both lecture and practical work to prepare students for hands-on experiments in an effective way. Physics Virtual Lab (PVL) contains the 3D virtual experiments, which are included in the physics curriculum such as mechanics, atomic and quantum physics, electricity and magnetism, and molecular physics. In each experiment users are able to change the initial parameters of the experiment and perform it several times with minimum efforts. PVL also provides to users theoretical material which includes useful information about the goal and step-by-step procedure of the experiment [141]. Gunawan et al. [142] conducted a study to examine the effect of the use of the virtual physics lab on problem-solving ability of students pertaining to the concept of electricity. In this study two groups were compared: experimental group and control group. Experimental group studied using an electric virtual laboratory, while the control group studied conventionally. It was found that problem-solving ability of experimental group is higher than the control group. Arista and Kuswanto [143] developed an Android application called Virtual Physics Lab (ViPhyLab) for the improvement of students' learning independence and conceptual understanding. The system is also equipped with exercise items where students could evaluate the concepts they learnt in ViPhyLab. PhysLab [144] is a 3D video game technology which is suitable for advanced level physics courses in secondary schools. In PhysLab two things to have confidence in students for doing experiments. First, it expresses the underlying mathematical models of the physics experiment. Second, data generated by the simulation must agree with real-world experimental data for validation. It can be used by teachers during development of concepts and theory to format the students' mental models for actual activities in physics.

Ghoniem et al. [145] have developed an intelligent object-oriented 3D simulation system for conducting Physics experiments. The system consists intelligent object-oriented paradigm for outlining the software architecture of the 3D simulations in which 3D object is available for reusing in multiple portions of the application. The Water Cycle in Nature is an interactive system that combines both virtual laboratory (VL) and virtual reality (VR) technologies [146]. In this system the contents are integrated into the VL-VR environment. The system provides a description of the natural water cycle definitions, such as vaporization, evaporation, boiling and condensation and examples of these phenomena occurring in nature. The user is also guided by both audio and textual instructions. Labster [147] is an online 3D environment for engaging students remotely [148]. In Labster students can simulate physics experiments about basic electricity, electrical resistance, Newton's laws of motion, vectors and scalars, principles of spring and masses etc. Similarly, PraxiLabs [149] is an online 3D virtual physics lab that consists various physics experiments such as Boyle's law, specific heat of solids, properties of matter, mechanics etc. In PraxiLabs students are facilitated by visual information both in English and Arabic. In addition, it also provides students' learning management system to assess their learning process [150]. For online physics teaching a virtual lab has been developed for Moroccan students. It is based on open-source learning (Moodle) platform which facilitate users to create their personalized learning environments. In this system user first consults the theoretical course reminder, then contributes to a multiple choice question (MCQ) formative assessment. Next, the user watches the laboratory video of each activity, then displays the simulation report and the operating mode. After that, the user simulate an experiment according to the experimental protocol. Finally, the user does the assessment activity and submits it to the concern tutor.

The system provides 12 virtual practical activities of physics, such activities are resistance measurement in electricity, prism, simple pendulum calorimetry, static and dynamic study of spring etc [151]. The analysis of the existing 3D-IPLs is summarized in Table 6.

**Table 6.** Analysis of the existing interactive physics laboratories in 3D.

| S. No | Reference | Year | Interaction Devices | Guidance (Assistance) | Venue | Remarks |
|---|---|---|---|---|---|---|
| 1 | Inoue et al. [138] | 2005 | Phantom Omni | No | Conference | It is used for pulley experiments in physics to know about balance of force, work and motion equations. |
| 2 | Karagöz et al. [139] | 2010 | Mouse + Keyboard | No | Journal | It is used for the simulation of electric and electronics circuits-based experiments on secondary and higher education level physics curricula. |
| 3 | Karingula et al. [140] | 2013 | Mouse + Keyboard | Virtual Instructors | Conference | It is used for the simulation of high school level physics experiments such as spring mass system, gravity drop, pendulums, two-masses pulley-based experiments, and vibration modes of string etc. |
| 4 | Daineko et al. [141] | 2017 | Mouse + Keyboard | Textual-based Guidance | Journal | The system is used for the physics curriculum such as mechanics, atomic and quantum physics, electricity and magnetism, and molecular physics. |
| 5 | Gunawan et al. [142] | 2017 | Mouse + Keyboard | None | Journal | It is suitable for the problem-solving ability of students to the concept of electricity. |
| 6 | Arista and Kuswanto [143] | 2018 | Mouse + Keyboard | None | Journal | In this system students could understand the concept of rotational dynamics materials such as rotation of a wheel. |
| 7 | Colin and Ruth [144] | 2019 | Mouse + Keyboard | None | Journal | The system is used to motivate the students for such physics experiments including oscillations of mass, electric and magnetic field, momentum etc. |
| 8 | Ghoniem et al. [145] | 2020 | Mouse + Keyboard | None | Journal | The system consists intelligent object-oriented paradigm in 3D object such as magnet, motor, lamp, etc. can be reused in multiple portions of the application. |
| 9 | Diana et al. [146] | 2020 | Mouse + Keyboard | Audio and Textual Instructions | Journal | In this system the contents are integrated into the VL-VR environment to explore the natural water cycle process. |
| 10 | Zourmpakis et al. [148] | 2022 | Mouse + Keyboard | None | Journal | In Labster students can remotely simulate the physics experiments such as electricity, electrical resistance, Newton's laws of motion, vectors and scalars, principles of spring and masses etc. |
| 11 | Ibrahem et al. [150] | 2022 | Mouse + Keyboard | Textual-based Guidance | Journal | It is used to simulate experiments such as specific heat of solids, Boyle's law, properties of matter, mechanics etc. |
| 12 | Kharki et al. [151] | 2022 | Mouse + Keyboard | Visual-based Guidance | Journal | The system provides a Moodle-based platform for online physics teaching. |

In this section, we elaborated previous works about ISLs. The existing ISLs are playing a supportive role in educational institutions to train the students for hands-on lab works. However, the existing ISLs have variant constraints (see Section 4 which motivate us for further improvement of ISLs.

### 3.4. Comparative Analysis of Interactive Laboratories Both in 2D and 3D

This subsection describes the systematic observations about similarities and differences between 2D and 3D ISLs. Further, in Table 7 pros and cons of ISLs both in 2D and 3D are also discussed.

**Table 7.** Comparison between 2D and 3D interactive laboratories.

| S. No | 2D ISLs | 3D ISLs |
|---|---|---|
| 1 | 2D ISLs provide only two degrees of freedom (2DoF) [56]. | 3D ISLs allow three degrees of freedom (3DoF) to six degrees of freedom (6DoF) [80]. |
| 2 | 2D ISLs are less realistic environments and hence provide low immersion [14]. | 3D ISLs are more realistic and immersive environments where users feel as though they are in physical labs [14]. |
| 3 | In 2D ISLs to search out any instrument/object is easier due to the smaller saccade [152]. | In 3D ISLs searching for instrument/object is not easy due to a larger saccade [153]. |
| 4 | In 2D ISLs eye movements are typically smaller and are not accompanied by head movements and users feel less fatigue [14,152]. | In 3D ISLs eye movements are energetically costly due to head movements and users feel more fatigue [14,153]. |
| 5 | Online 2D ISLs require low internet speed for simulating biological experiments remotely [154]. | Online 3D ISLs require high bandwidth for simulating experiments [154]. |
| 6 | In 2D ISLs interaction with apparatuses is simple and users can directly select the require instrument/object [155]. | In 3D ISLs interaction is complex as navigation is also required prior to select and/or manipulate an instrument/object, i.e., glassware (test tubes and slides etc.) [80]. |
| 7 | 2D ISLs use mouse and keyboard for interactions [14]. | 3D ISLs use the most advance 3D trackers for interaction [14,80]. |
| 8 | 2D ISLs do not give spatial depth to the objects/models/instruments due to which they are less realistic [156]. | 3D ISLs allow spatial depth to the chemicals/models/instruments on the screen and deliver stronger feeling of reality [156]. |

We analyzed the similarities and differences between 2D and 3D ISLs and also discussed their pros and cons. In science education both ISLs are used constructively as learning tools and are very useful to enhance students' practical skills for physical lab works. However, on the basis of realism, immersion and spatial depth of chemicals/models/instruments there are vast differences between 2D and 3D ISLs. In a physical science laboratory/room users are able to move freely from one position to another, touch any objects (bio-objects/chemical objects) and other glass wares/apparatuses, view the objects from different sides, and manipulate the objects according to the experimental task. Similarly, 3D ISLs also provide an environment where users are able to navigate and select any objects like in a physical science laboratory. Therefore, 3D ISLs provide more real scenes as compared to 2D ISLs and users can achieve full immersion [157]. For achieving full immersion in 3D ISLs different technologies are used, such as HMD (Head Mounted Display) [158], CAVE (Cave Automatic Virtual Environment) [73] and Oculus VR Headsets [159]. Through these immersive technologies users can feel their physical presence and achieve fully-immersion in virtual environment. These technologies are also useful for learners' motivation and can also make the 3D ISLs more interesting for the simulation of experiments [73]. However, these technologies cannot be used widely due to their high price. To deliver better subject matter 3D ISLs provide stronger feeling of reality than 2D ISLs. According to the results, 3D ISLs allow users to interact with objects/models/instruments and contexts in a more naturalistic way [160]. However, in some other cases such as mathematics education [161] and learning alphabets [162] etc. 2D environments are suitable where physical movement is not necessary.

## 4. Challenges in Existing Interactive Labs

The existing ISLs are efficient systems for familiarizing the learners with their lab works and can resolve the issues encountered by different centers of learning/institutions in lab activities. However, the existing ISLs have variant issues and constraints [14,89]. For example, in existing ISLs if a student has to simulate a specific experiment, in a interactive lab that contains more objects (bio-objects/chemical objects) and other glass wares/apparatuses etc. These chemicals and glassware/apparatuses are distributed in different locations (tables, shelves, etc.), from where the bio-objects/chemical objects and glass wares/apparatuses are chosen for the current experiment puts more cognitive load/mental burden on the student [14]. In addition, in a learning environment students' performance can be affected due to large amount of informative materials which leads to cognitive load/mental burden on them [89,163]. In the current context, a mental/cognitive power can be defined as the strength of mental power to perform an activity and handle its related information [164]. The cognitive load is the amount of mental processing required to identify an object (bio-objects/chemical objects, and other glass wares/apparatuses etc.) in an interactive lab will increase if it contains a large number of chemicals, glassware and apparatuses [14]. The increased cognitive load badly affects students' performance. Different cognitive aids such as audio, visual or haptic-based instructions are used for reducing students' cognitive load. Actually these cognitive aids consist of instructions in textual or sound forms and display on the screen to guide the students what to do, for example, what bio-objects/chemical objects to use, what glass ware/apparatuses to use, and in which sequence or order. The student follows these instructions step-by-step while performing the experiments. However, all the students cannot interpret these instructions in a correct manner because they read the textual aids and hear audio aids but cannot understand correctly the simulation of an experimental task in reality [20,165]. Furthermore, none of the existing virtual laboratories have used such cognitive aids to help students carry out an experiment in the correct manner with high performance and improved learning [14]. In majority cases, the experiments in science subjects (such as biology, chemistry and physics) consist of predefined stepwise procedures including sequence of tasks and their scheduling (time) are important for the experiment according to their correct procedures. Performing of experiments are the difficult tasks for students, because during experiments students can often forget one or more experimental steps or alter the sequence, which causes undesired results. In this context, the aids (instructions) may beneficial for students in the conduction of their experiments [20,90]. Therefore, it requires the provision of such cognitive aids through which users can easily interpret the actual procedure of an experiment(s).

In science education, experimental tasks are among the difficult tasks when transferring each course of action about the experimental tasks to learners/students [14,90]. Many issues happen during learning of experimental tasks which affect their mental models and learning skill [20]. In these issues one of the key issue is the boredom of students where students do not take any interest [166,167]. Therefore, sometimes students become bored when they learn or perform any experimental tasks [14,168]. In the existing ISLs, there is no source to remove boredom from students during simulation of experiments [14].

Overall the existing ISLs are used for the simulation of pre-defined experiments, procedure or safety procedures and cannot be adapted according to the students' level [14]. In the existing ISLs only the developers/programmers are the authentic persons to update their systems [151]. Users/instructors cannot add a new experiment, objects such as chemicals, cells, skeletons, human models and other glass wares/apparatuses or update its properties by adding new information for their students, is also a research question.

In the light of these constraints, we conducted a subjective study and discussed these issues with the field experts to propose novel solutions (see next Section 5).

## 5. Field Survey and Research Significance

This section presents a subjective survey from the field experts from which we have proposed the possible solutions of the actual problems. We used the existing PbVCL [90]

to know the actual problems such as hurdles in use and its practical implementation and possible solutions of these problems. In connection with the constraints previously mentioned, we arranged a subjective survey by interviewing the field experts (i.e., chemistry, biology and physics). Figure 3 shows the sequence of our study which is divided into different phases.

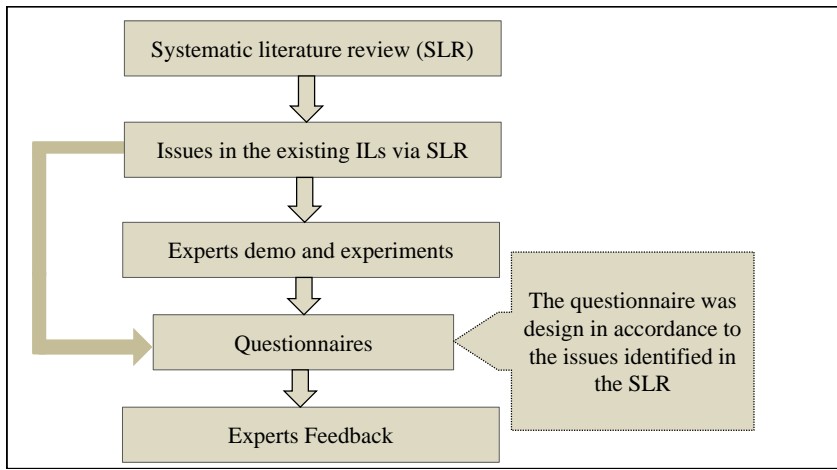

**Figure 3.** Different phases of our study and their relationship.

### 5.1. Interviewing from Field Experts

To investigate the key issues in the existing ISLs, and difficulties in its practical implementation and use, we arranged a subjective survey by interviewing the field experts.

In this phase, we arranged interviews with science subjects' experts (both male and female experts) to obtain their suggestions for possible improvements in our existing Pb-VCL [90]. The PbVCL [90] is shown in Figure 4. In this interview twenty-four experts participated from different institutions such as secondary school, higher secondary school and university level (i.e., nine of chemistry, eight of biology and seven of physics teachers). These experts were informed before the interviews about the aim of the survey and requested for appointments in their office hours. The average duration of each interview with experts was 35 to 40 min. The interview was consisted of four steps. The first step addressed the teachers' backgrounds, that is, their prior education, their experiences and their working context in a generic sense. In the second step, they were briefed with the help of 10 to 15 min demonstration about the basic functions of our existing PbVCL [90]. In the third step, they performed different tasks such as navigation, selection, and manipulation of chemicals/equipment in our existing PbVCL [90]. We also briefed them about the functions of different aids, i.e., different combination of arrow, audio and textual-based aids. In the last step of the interview, we focused on the implementation of an innovative ideas in the concerning subject. In particular, the teachers' ideas and expectations concerning learning efficiency, ease of use, ease of chemicals' search and the suitability for the experiments. We asked their opinions regarding the key issues in the practical implementation and use other current PbVCL [90]. The second series of the interview protocol are listed in Table 8. During questionnaire interview the teachers answered each question on a scale of four options except question 5 which considers only one option.

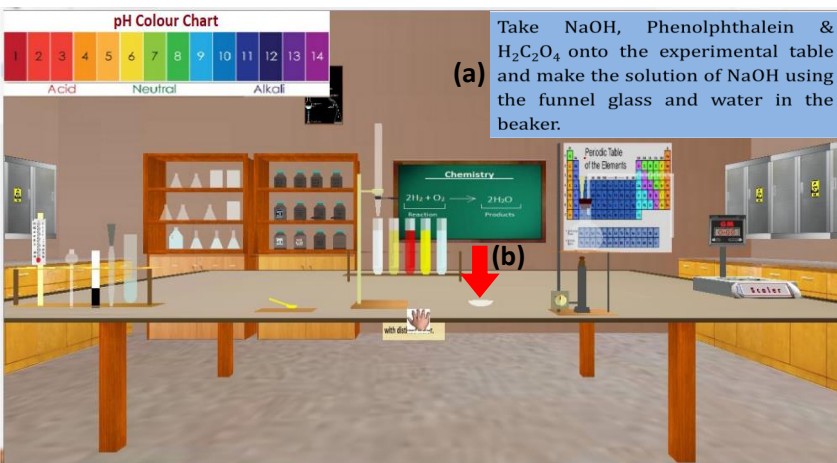

**Figure 4.** The existing virtual lab with aids; (**a**) represents textual instructions and (**b**) represents arrow [90].

**Table 8.** Questionnaire interview from field experts.

| Q.No | Questions |
|---|---|
| **Q1** | What is the best guidance technique to make the interactive lab easy in understanding? (a) Arrow + Textual aids (b) Arrow + Audio aids (c) Textual + Audio aids (d) Other suggestions |
| **Q2** | What makes the experimental steps easy for students to perform in the interactive lab? (a) Arrow + Textual aids (b) Arrow + Audio aids (c) Textual + Audio aids (d) Other suggestions |
| **Q3** | What is the best guidance technique to make the interactive lab easy in searching and selection of the chemicals or glass wares during the current experiment? (a) Arrow + Textual aids (b) Arrow + Audio aids (c) Textual + Audio aids (d) Other suggestions |
| **Q4** | Rate the degree of importance (1–5) for the following usability metrics, where 1 means less important and 5 means very important. (a) Minimal cognitive load (b) Easiness in use (c) Use of technical guidance (d) Integration of the modules |
| **Q5** | Give further suggestion for removing of boredom from students and practical implementation & use of the PbVCL [90] to handle the key issues. |

*5.2. Feedback from Experts*

Table 9 summarizes the feedback of each expert. Firstly, they investigated our existing PbVCL [90], after that their responses were collected using a questionnaire interview. The first question, which is related to the easiness in understanding of the interactive lab during the performance of experiments. For the first question, 8.6% of the experts selected "Arrow + Textual Aids" option, 13.1% selected the "Arrow + Audio Aids" option, 30.4% selected the "Textual + Audio Aids" option and 47.8% selected the "Other Suggestions" option. In each question the "Other Suggestions" option is used to find an innovative idea from each expert. This means that most of the experts provided their own ideas which will assist students by an easy cues in the understanding of interactive lab during experimental setup. Similarly, the second question, which is related to the easiness in performing of experimental steps according to the correct procedures by using aids. For the second question, 52.1% of the experts selected the "Other Suggestions" option. Like a first question here also most of the experts provided their own ideas for assistance the students during performing of experimental steps. The third question, which is related to the searching/selection of the chemicals and glass wares according to the experimental steps during the selected experiment. For the third question, 60.9% of the experts selected the "Other Suggestions" option. Considering the option "Other Suggestions" from the question 3 most of the experts suggested the use of only arrow aid for searching/selection of the chemicals and glass wares according to the experimental steps. Similarly, for the fourth question, which is related to rate the degree of importance for the usability metrics to validate the issues in the existing PbVCL [90]. For the first option which is related to the

minimal cognitive load, 56.5% of the experts selected the "very Important" option. For the second option which is related to the easiness in performing of experimental steps, 52.1% of the experts selected the "very Important" option. For the third option which is related to the use of technical guidance in performing of experimental steps, 56.5% of the experts selected the "very Important" option. For the last option, which is related to the integration of modules, 47.9% of the experts selected the "very Important" option. This means that on behalf of question 4 the experts validated the issues in the existing PbVCL [90] via rating the degree of importance for the usability metrics. On behalf of the question 5 and considering the option "Other Suggestions" from the above questions, we reached to the following proposed ideas which are suggested from field experts.

**Table 9.** Experts' Responses (Participants = 23). Each cell indicates the %age value.

| Questions | Experts' Responses | | | |
|---|---|---|---|---|
| | **Arrow + Textual Aids** | **Arrow + Audio Aids** | **Textual + Audio Aids** | **Other Suggestions** |
| **Q1: Easiness in understanding** | 8.6 | 13.1 | 30.4 | 47.8 |
| **Q2: Easiness in performing** | 13.1 | 8.7 | 26.1 | 52.1 |
| **Q3: Easiness in searching** | 4.3 | 17.4 | 17.4 | 60.9 |
| **Q4: Usability metrics** | **Less Important (1)** | **2** | **3** | **Very Important (4)** |
| **Minimal cognitive load** | 8.6 | 13.1 | 21.8 | 56.5 |
| **Easiness in use** | 13.1 | 8.6 | 26.1 | 52.1 |
| **Use of technical guidance** | 14.3 | 9.5 | 19.7 | 56.5 |
| **Integration of the modules** | 8.6 | 8.6 | 35.0 | 47.8 |

## 6. Discussion and Future Directions

This section presents a general discussion on our findings with a summary of the study and constraints in the existing ISLs, and the novel solutions for the development of future ISLs. This section also describes the limitations of our research study.

### 6.1. Summary of the Study

For this study, we reviewed different types of existing ISLs including 2D and 3D in science education. We followed the systematic literature review (SLR) methodology [42] for article searching, selection, and quality assessments. We selected 86 articles with 34 in chemistry (see Tables 1 and 2), 27 in biology (see Tables 3 and 4), and 25 in physics (see Tables 5 and 6) after final selection using SLR [42] and classified these articles into different categories. Each article is selected after briefly studying basis on different information extraction, including category of the article, key idea, evaluation criterion, and its strengths and weaknesses. Figure 5 shows a chart of the articles' quality assessment.

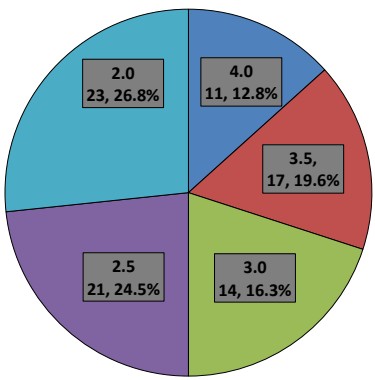

**Figure 5.** Pie chart showing percentage values of article in each category of the quality assessment (i.e., Quality assessment score, number of articles, and percentage value).

### 6.2. Summary of Challenges

The existing ISLs are efficient systems for familiarizing the learners with their lab works and can resolve the issues encountered by different institutions in lab activities. However, the existing ISLs have variant issues and constraints [14,89] as already discussed in Section 4, these issues and constraints are the following:

- In the existing ISLs students' performance can be affected due to large amount of informative materials which leads to cognitive load/mental burden on them [89,163].
- In the existing ISLs students' cannot interpret the cognitive aids in a correct manner during simulation of experiments [20,165].
- Sometime students become bore when they learn or perform experimental tasks [167,168]. There is no source in the existing ISLs to remove boredom from students during simulation of experiments [14].
- Overall the existing ISLs are used for the simulation of pre-defined experiments which provide static information and cannot be updated according to the students' level [14].

In the light of the above discussed constraints with field experts, the following novel solutions for the improvement of forthcoming ISLs.

### 6.3. Cognitive Load and Adaptive Aids

In real laboratories, students normally carry out the lab works according to their teacher's instructions without any hurdles and cognitive load/mental burden [14]. Therefore, some of the existing ISLs have used textual and audio-based aids to carry out an experiment as discussed in Section 4. The main issue to perform an experiment in laboratory is to execute the whole procedure according to the correct course of action [20,89]. Therefore, we proposed a novel solution to provide step-by-step adaptive aids. The adaptive aids approach will consist of an arrow and animated guidance which will be displayed autonomously according to the actual experimental tasks. In adaptive aids, an arrow will be used in the selection of objects/chemicals while animation will display on the screen the procedure of the actual experimental task of the current experiment. For example, when a student searches any objects and glassware, the system will use arrow aid to assist him/her in the selection of correct items, and the animation will be used to display the actual experimental task to perform the current step in a correct manner. An animation is the illusion of movements created by showing a series of images about the actual steps and tasks in each experimental step. The student will follow the instructions in image forms about the actual task and will perform the experiments. It will help students to take interest and perform the experiment more easily without taking any help from the expert. Through adaptive aids it will trace the user's actions to know about task's completion and once it detects that the student has completed all of the actual task in the current step, then the system will detect the next step to display such aids (arrow aid or animated) for students to perform it, and so on. This mechanism will be very useful to minimize the mental burden on students/users. Furthermore, the students can complete the experiment independently according to the correct course of action and such aids may fulfill the need of an instructor.

### 6.4. Boredom and Background Calm Music

Boredom is also an issue in the existing ISLs [14]. Most of the gamified virtual environments facilitate users by background calm music, so that, to make them enthusiastic for doing such gamified task without any boredom [169,170]. Therefore, it is also suitable to include calm background music in virtual science labs for students' natural curiosity and removing such boredom from them and to make them enthusiastic for such learning or performing science experiments.

### 6.5. Dynamic ISLs

The existing ISLs are static and only provide the simulation of predefined chemical experiments/reactions, procedures, or safety measures [14]. A user/instructor cannot



register a new experiment, objects (cell, skeleton, chemical, glassware, apparatus etc.), or add something new to its properties [151]. Therefore, the proposition and development of a dynamic framework, which allows the users/instructors to add a new chemical reaction by registering its objects, chemical items, apparatuses, and procedure according to their students' class level. The framework will use this information in the simulation of an experiment.

### 6.6. Limitations of This Study

The following are the main limitations of our survey:

- Although we have used SLR in the searching, inclusion and exclusion of the articles. However, it is possible that due to too many articles related to the ISLs, we might have excluded some relevant articles.
- This research study is consisted only those articles which are written in English language. Therefore, articles which are written in other languages would enhance further the research outcomes.
- This research study is related to the ISLs in science education (i.e., chemistry, biology and physics). However, in subjective study with field experts to suggest novel solution, we used only the existing virtual chemistry lab [90]. Therefore, virtual lab which are related to other domains such as virtual physics and biological labs would enhance further novel solutions for future ISLs.

However, these limitations do not affect the value of this research study; in fact, our findings mould the source for further development of interactive science labs to overcome the current research gaps.

### 7. Conclusions

In this paper we presented a comparative review of interactive labs in science education. We focused on interactive science labs in the three major subjects of science such as chemistry, biology and physics. In the first phase, we studied the existing state-of-the art interactive science labs including ICLs, IBLs and IPLs. In total, we selected 86 articles via SLR methodology. We analyzed these articles critically with their pros and cons. We also probed to highlight some key constraints in the existing ISLs. In science education, the existing ISLs are efficient systems for familiarizing the learners with their lab works. However, The existing ISLs have various limitations and constraints including cognitive load/mental burden, guidance/cognitive aids, boredom of students and restriction on users in the addition of new information or experiments. In the second phase, considering the limitations and constraints, we conducted a subjective study with field experts by investigating one of our existing virtual lab about the practical implementation and to find out novel solutions for the future ISLs. This study, while analyzing the existing literature, highlights some research issues that would be helpful for the development of future interactive science labs.

**Author Contributions:** Conceptualization, N.A.; methodology, N.A., S.U. and D.K.; validation, D.K. and S.U.; writing—original draft preparation, N.A.; writing—review and editing, N.A., S.U. and D.K.; supervision, S.U. All authors have read and agreed to the published version of the manuscript.

**Funding:** There was no funding to support this research.

**Conflicts of Interest:** We have no conflict of interest with any organization in this case.

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
