# Peer review of "Interactive Laboratories for Science Education: A Subjective Study and Systematic Literature Review"

_mti, doi:10.3390/mti6100085_

Round 1

Reviewer 1 Report

We must thank the authors for this work. The scientific level is correct.

I would like to highlight some aspects of improvement.

This study covers a wide area: chemistry, biology and physics. Perhaps it would be appropriate to better differentiate among these areas.

The number of sample articles is not very high, with some places showing 89, 90 and 86 on another. Other cases, the research on scientific literature collects more than 200 records, in this sense, it is very important to collect the reviews of previous similar research, e.g.n this article DOI: 10.1039/d1rp00317h appears the research until 2020 in the field of chemistry and virtual reality.

It is also not very clear if the analysis focuses only on the years 2020 to 2022, since there is many references to other previous articles.

In my opinion, it is also necessarys to show some reference to laboratories supported by augmented reality technology, since there is no reference about this technology. In the same way, it would be good to take the previous literature analyzed, e.g. DOI: 10.1021/acs.jcim.1c01255

Methodologically, the review part of the scientific literature is clear. The methodology of the part of the interview with the experts is not clear, nor is the analysis of the validity of these interviews and their limitations.

In my opinion, the results should be analyzed in a broader discussion section and writing down very clear contributions to the academic community and differentiated by areas.

Formally it would be necessary to review the references since there are some errors, among others:

There is  referenced material with erroneous content, e.g. Netlist 

Stone [70] Mistake.... Limniou, M.; Roberts, D.; Papadopoulos, N.

Karingula, M.V.K.; El-Mounayri, H.A. Assessment of Virtual Physics Lab (VPL) in summer course for pre-college preparation. age, 23, 1

NOT Working. Biology Labs Online https://www.sciencecourseware.org/BLOL/

References from non-scientific areas: Vooys, P. CellCraft Game Exploration.http://petervooys.weebly.com/, 2016. 

References to Software: Crocodile Simulation Software.

JEL: JHKSOFT electricity lab.

References to Companies: Edison4: DesignSoft Company.https://www.designsoftware.com/home/English/. 

Blogs: James, P. This is the beginning of VR Education, and it will only get better. 2014   ???

ETC.

In this sense, scientific articles must be differentiated from other less rigorous sources.

Reviewer 2 Report

The authors are to be commended for a comprehensive, very readable review of interactive science labs. Here are just a few suggestions to the authors:

1. Since the focus was on chemistry, biology, and physics labs, the title should reflect the slightly more narrow scope than "science education".

2. The search criteria are well defined, but the quality assessment of articles to narrow down to the final set is still a little unclear to a reader. For instance, why was 3 a cutoff for citations, or how exactly were characteristics like innovativeness or relevance assessed?

3. The authors do a nice job describing all of the studied articles but there is not much in the way of summarization in each of the sections.

4. Summarization is not done across science topics (biology, chemistry, physics). If there are no significant differences then the authors might state so.

5. Summarization is done between 2D and 3D labs. This is welcome, but one aspect the authors might consider more explicitly is when the different formats are beneficial or not. For instance, it is not always the case that physical movement is needed, so 2D environments in those cases may be as or more appropriate than 3D environments.

6. This reviewer is not sure the separate work with experts is needed for this manuscript. It is focused on one specific lab and the questions asked of experts do not tie specifically to research questions that drove the original research. The authors may do a qualitative analysis of experts' opinions and create a separate paper.

7. The authors' solutions for labs are welcome but bring up a host of questions because they tie into vast literature. For instance, adaptive aids have been heavily used. Further, dynamic labs are desirable but their creation adds a lot of complexity that is not addressed here.

8. The authors may wish to have an editor look over the paper for minor edits such as duplicative phrases, slight grammatical fixes (e.g., "become bore" instead of "become bored"), and missing punctuation such as some closing parentheses.

Overall, though, a well-crafted paper and well-done work.

Reviewer 3 Report

Dear authors

It's a very interesting article with two different methodologies. There are several comments to enrich the manuscript and to improve the quality of this research

-The introduction is so short. It is advised to enrich it by more paragraphs.

- It’s preferable to write the introduction as paragraphs, it’s written as subtitles with bullets.

- It’s beautiful to have a paper organization but it’s not a familiar part of research structure.

- The authors are advised to present the exclusion criteria.

- A table of systematic review themes should be shown before presenting the analysis results.

- It’s recommended to have a table to present the interview themes if there.

- I wonder if authors could link results of the systematic review and the subjective study.    

Round 2

Reviewer 1 Report

I think the article has improved. Congratulations.

However, again I see a problem of covering a field and a very broad analysis... I observe the article analyzes information from very varied sources and with little rigor in some cases. In my opinion, research published in high-impact journals should not be mixed with the websites of companies or entities with no scientific relevance.

For instance in Table 3. Analysis of the existing interactive biological laboratories in 2D...    scientific articles are combined with other less rigorous sources

For example

a  Proceeding

96. Slator, B.M. Intelligent tutors in virtual worlds. Proceedings of the 8th International Conference on Intelligent Systems (ICIS-99). June, 1999, pp. 24–26.

-This paper originally appeared as Slator, Brian M. (1999). Intelligent Tutors in Virtual Worlds. Proceedings of the 8th International Conference on Intelligent Systems (ICIS-99). Denver, CO. June 24-26, pp. 124-127. (http://cs.ndsu.edu/~slator/html/abstracts/icis-99-final.html)

a Biology Labs Online

97. BLO. Biology Labs Online.https://www.sciencecourseware.org/BiologyLabsOnline/. Accessed: 30-08-2022

… it is not a Journal

Rajaram and Ivan… you write their NAMES –it should be SURNAMES-

99. Subramanian, R.; Marsic, I. ViBE: Virtual biology experiments. Proceedings of the 10th international conference on World Wide Web, 2001, pp. 316–325.

...a Virtual WEBSITE

105.Virtual Urchin.https://depts.washington.edu/vurchin/. Accessed: 11-04-2022.

(With 106. Haverkort-Yeh,...)

They are doubts that arise from Table 2... and so on with the other tables...

Thus, you say “for this research study, the literature related to the use of ISLs in science education (i.e., chemistry, biology and physics) published as full length articles both in scientific journals and conferences during 1999 to 2022 were studied.” That is why I encourage you to clearly differentiate in this article when analyzing scientific research from the sources that you indicate “ACM, Elsevier, IEEE, MDPI, Springer Link, and Taylor & Francis”. As I indicated in my first review, perhaps "scientific articles should be differentiated from other less rigorous sources".

(On the other hand, the use of references that are more than 20 years old is very complicated in the field of technology.)

Reviewer 3 Report

Thanks. The authors did a good job.
